# Intelligent Dynamic Real-Time Spectrum Resource Management for Industrial IoT in Edge Computing

**DOI:** 10.3390/s21237902

**Published:** 2021-11-26

**Authors:** Deok-Won Yun, Won-Cheol Lee

**Affiliations:** 1Department of Electronic Engineering, Soongsil University, Seoul 06978, Korea; dhtor@naver.com; 2School of Electronic Engineering, Soongsil University, Seoul 06978, Korea

**Keywords:** spectrum management, data mining, data preprocessing, artificial neural network, case-based reasoning, interference analysis, cognitive radio

## Abstract

Intelligent dynamic spectrum resource management, which is based on vast amounts of sensing data from industrial IoT in the space–time and frequency domains, uses optimization algorithm-based decisions to minimize levels of interference, such as energy consumption, power control, idle channel allocation, time slot allocation, and spectrum handoff. However, these techniques make it difficult to allocate resources quickly and waste valuable solution information that is optimized according to the evolution of spectrum states in the space–time and frequency domains. Therefore, in this paper, we propose the implementation of intelligent dynamic real-time spectrum resource management through the application of data mining and case-based reasoning, which reduces the complexity of existing intelligent dynamic spectrum resource management and enables efficient real-time resource allocation. In this case, data mining and case-based reasoning analyze the activity patterns of incumbent users using vast amounts of sensing data from industrial IoT and enable rapid resource allocation, making use of case DB classified by case. In this study, we confirmed a number of optimization engine operations and spectrum resource management capabilities (spectrum handoff, handoff latency, energy consumption, and link maintenance) to prove the effectiveness of the proposed intelligent dynamic real-time spectrum resource management. These indicators prove that it is possible to minimize the complexity of existing intelligent dynamic spectrum resource management and maintain efficient real-time resource allocation and reliable communication; also, the above findings confirm that our method can achieve a superior performance to that of existing spectrum resource management techniques.

## 1. Introduction

Based on the recent development of the Internet of Things (IoT), a paradigm shift is rapidly occurring in computing technology, creating the ability to effectively process large amounts of data generated by various industrial IoT devices [1]. According to the International Data Corporation, by 2025 there will be 44.1 billion internet-connected IoT devices generating close to 79.4 zettabytes of data [2], representing a significant departure from major end–end smart device terminals to the machines, sensors, and cameras that constitute the IoT. Gartner also predicted that approximately 10% of enterprise-generated data is already being processed outside of centralized data centers or the cloud, and that this number will reach approximately 75% by 2025 [3]. This suggests that cloud computing with a centralized computing structure has limitations in terms of accommodating Industrial IoT services requiring real-time processing, such as smart factories, smart farms, and autonomous vehicles. Therefore, due to the risk of overloading cloud servers based on the exponentially increasing data volume and network traffic, the transition of computing from the current remote clouds to a new paradigm of network edges within the radio access network is expected to gain popularity [4,5,6].

Edge computing is a distributed computing pattern that brings computation and data storage closer to the sources of data. This means when computing services are processed in a location close to the terminal user device, the user can receive faster and more stable services and can benefit from flexible hybrid cloud computing. This method is particularly effective for overcoming the problem of long latency, an issue that must be overcome to achieve real-time data processing and decision-making in smart factories using the IoT. It also supports real-time decision-making through sensing data collection and analysis. however as shown in Table 1, The demand for access to finite spectrum resource is increasing, as new technologies allow a variety of applications to make use of a broader range of frequency bands [7,8,9,10,11,12].

This in turn results frequent spectrum handoffs in dynamic spectrum environments that require machine type device (MTD) operation in real time, which may lead to serious performance degradation owing to interference with other homogeneous networks. Therefore, dynamic optimization according to time-varying parameters, such as channel state information, is preferable.

To achieve this, the previously studied intelligent dynamic spectrum resource management [13] constitutes an infrastructure-based WRAN in which a base station utilizing the TV White Space (TVWS) band controls IoT devices in a suburban area with limited broadband access support according to the IEEE 802.22 WRAN standard [14,15,16]. Each base station forms a single wireless network and has a learning, reasoning, and optimization engine structure made possible through database access, enabling dynamic spectrum resource management for coexistence between cells of the same network type existing in adjacent channels and locations. First, to facilitate timely handoffs to idle channels, the learning engine generates a list of backup channels through characterization, spectrum determination, and classification based on historic data representing the statistical information of the existing local network. A reasoning engine then uses the backup channel list to perform a spectrum handoff operation to another idle channel, enabling continuous communication without interfering with incumbent users. At this time, when interference is affected in incumbent users existing in adjacent channels and locations, transmission parameters that can coexist between neighboring networks are dynamically reconstructed through an optimization engine using interference analysis and genetic algorithms.

However, decision making based on the optimization algorithm increases computation time and complexity, making it difficult to rapidly allocate resources and wasting valuable solution information optimized according to the evolution of the spectrum state between the space–time and frequency domains [17,18]. Therefore, case-based reasoning is used in this study to reduce the complexity of intelligent dynamic spectrum resource management and achieve rapid resource allocation. However, if misclassified cases exist due to irregular patterns, system performance may deteriorate due to incorrect prediction; therefore, creating an accurate learning algorithm for case-specific data is imperative.

In this paper, we propose intelligent dynamic real-time spectrum resource management using data mining and case-based reasoning, as shown in Figure 1. The intelligent dynamic real-time spectrum resource management structure is largely divided into a CR Master and CR User. First, CR Master, which applies data mining and case-based reasoning, analyzes the activity patterns of existing users through data pre-processing and machine learning based on historyl data stored in the database and classifies them by case. In addition, when a new case is input, similarity verification is performed using the classified case DB. The CR User performs solution reuse and maintenance and update operations in case-based reasoning. At this time, the solution means the optimized transmission parameters and, depending on the suitability evaluation result achieved for the solution, a new solution is derived through solution retain or an optimization engine.

The intelligent dynamic real-time spectrum resource management operation is as follows. Initially, ① the sensing information of the Cognitive Radio (CR) User is transmitted to the database via the CR Master, which uses data mining at regular intervals to analyze the activity patterns of incumbent users from the historic data stored in the database. Data mining performs data collection, pre-processing, and machine learning operations for each reasoning cycle. When reasoning using case DB classified through data mining, ② CR Master determines whether or not to deliver solutions for cases similar to ones seen in the past through similarity verification. Subsequently, ③ in the idle channel reasoning process using the backup channel list, if the CR User affects influence of interference with the incumbent users existing in the adjacent channels or locations, the CR User operates the optimization engine according to the similarity verification result or ④ reuses the solution delivered from the CR Master. ⑤ The verification of the solution determines the operation of the optimization engine through Monte Carlo algorithm-based interference analysis, and ⑥ the optimized solution is updated in the case DB for the next reasoning.

The major contributions of this paper are as follows.

We developed an intelligent dynamic real-time spectrum resource management structure that combines data mining and case-based reasoning to minimize computational volume and complexity of existing spectrum resource management and enable rapid resource allocation within a limited time.We provide insight into spectrum management issues in various fields through intelligent dynamic real-time spectrum resource management.

The composition of this paper uses data collection, data preprocessing, data mining. and artificial neural networks (ANNs) to analyze the activity patterns of incumbent users from historic data and compares and analyzes the prediction accuracy and spectrum resource management performance based on the number of IoT devices involved.

## 2. Data Mining

In general, vast amounts of sensing data from industrial IoT deployed in poor environments are frequently generated by repetitive and irregular data owing to various transmission errors, sensor malfunctions, and cases of external interference, which can eventually lead to poor performance due to incorrect predictions being made in case-based reasoning [19,20]. Therefore, we propose the use of data collection, data preprocessing, and ANN to effectively improve prediction performance through the accurate pattern learning of sensed data [21].

### 2.1. Data Collection

In this study, the CR master learns traffic patterns for each average occupancy probability as well as backup channel selection mechanisms using historic data representing the statistical characteristics of incumbent users at regular periodic intervals from the database.

Here, the historic data representing the statistical characteristics of the incumbent user use the ON/OFF process model represented by the incumbent user activity model in the CR network (CRN) [22]. The historic data configuration using the ON/OFF process model has different exponential distribution characteristics in the *j*-th channel and calculates the occupancy probability $P_{on}^{j}$ for each channel using $\lambda_{on}^{j}$ and $\lambda_{off}^{j}$, which indicate the average values for ON and OFF state retention times.
(1)$$P_{on}^{j}=\frac{\lambda_{on}^{j}}{\lambda_{on}^{j}+\lambda_{off}^{j}}.$$
where j∈N, N=1,2,3,…,n.

At this time, the occupancy status of the incumbent users per channel until the reasoning period *T* is based on the following conditional expression according to the above occupancy probability value.
(2)$$C_{j}\left(T_{i}\right)=\left\{\begin{array}{cc} 0, & P_{on}^{j}\left(T_{i}\right)\leq k\\ 1, & P_{on}^{j}\left(T_{i}\right)>k \end{array}\right.$$
where i∈M, M=1,2,3,…,m.

Here, *k* corresponds to a random number between zero and one and is randomly generated for each channel until a given time *T*. At this time, if the random number *k* in the *j*-th channel at time $T_{i}$ is greater than the occupied state probability, it is judged that the incumbent user is highly likely to be ‘OFF’ and the function returns a zero. Conversely, if the occupied state probability is high, then the incumbent user in the *j*-th channel is likely to be ‘ON’ and the function returns a one.

Figure 2 shows historic data representing the status information of incumbent users from the past to the present using such conditional expressions.

### 2.2. Data Preprocessing

In this study, data preprocessing process consisting of data cleaning, data reduction, outlier detection, and data scaling are performed using the history data defined in the data collection stage as shown in Figure 3 [23,24].

Data cleaning aims to construct new characteristics based on linear combinations of existing variables. As a representative linear feature extraction technique, a statistical feature extraction method calculates statistics such as the average or median of raw data within a specific time range [25]. However, severe time series fluctuations can lead to data loss, so statistical data samples were reduced to categorical data in this study. The term ’categorical data’ refers here to the frequency of values belonging to the corresponding interval, dividing the range of statistical data values into several intervals of the same size. Through this transformation, this method can minimize information loss and data size, effectively reducing the amount of computation required when analyzing large-scale data.

Following on from the above method, an occupancy probability $P_{on}^{j}$ representing the statistical characteristics of the sensed data within the total *M* time slots per channel is calculated in this study.
(3)$$P_{on}^{j}=\frac{1}{M}\sum_{i=1}^{m}C_{j}\left(T_{i}\right)$$

At this time, the interval width Δx divided based on the specified number of intervals *L* within the categorical data set range (*min*, *max*) is defined as follows:(4)$$\Delta x=\frac{max-min}{L}$$

If $P_{on}^{j}$ exists in the range of ±0.5Δx based on the *k*-th median value $\chi_{k}$ among pieces of statistical data, it is defined as one; otherwise, it is zero. In this case, the frequency of the appearance of $H_{k}$ indicates a result value counted up to l−1 per interval.
(5)$$\chi_{k}=min+0.5\Delta x+k\Delta x$$
(6)$$\Delta_{k}\in \left(\chi_{k}-0.5\Delta x+0.5\Delta x\right)$$
(7)$$y_{k}\left(P_{on}^{j}\right)=\left\{\begin{array}{cc} 1, & P_{on}^{j}\in \Delta_{k}\\ 0, & elsewhere \end{array}\right.$$
(8)$$H_{k}=\sum_{k=0}^{l-1}y_{k}\left(P_{on}^{j}\right)$$
where k∈L, k=0,1,2,…,l−1.

In addition, repetitive and irregular data are highly likely to generate outliers during data preprocessing due to various transmission errors, sensor malfunctions, or types of external interference occurring during the data collection process. This causes the data analysis accuracy to decrease by excessively increasing the variance in estimating the population average of the data. To counteract this, outlier detection was used in the data reduction process. This method represents the characteristics of each channel by calculating raw data with which it is difficult to distinguish between outliers and noise as statistical data. Then, after the frequency of appearance is calculated by dividing several intervals within the range of statistical data values, the data points in the interval with the fewest observations are identified as outliers and converted to zero.

Finally, data scaling is used to maximize the classification and prediction accuracy using an ANN based on supervised learning. To accomplish this, discretization was used in a previous study, allowing for pattern classification according to data characteristics and enabling efficient searching to take place [26]. However, to classify patterns of categorical data with binary values of zero and one, it is necessary to have a categorical data set with a high resolution that is distinguishable from neighboring datasets with other characteristics. Therefore, we used min–max normalization to solve this problem.

Min–max normalization rescales the range of each component *x* in a categorical data set to a range between zero and one. The general formula for normalization is as follows [27]:(9)$$x^{\prime }=\frac{x-min}{max-min}$$

Here, *max* and *min* denote the categorical data set size and $x^{\prime }$ denotes the normalized value.

### 2.3. Machine Learning

The CR Master classifies the preprocessed historic data by case and determines whether or not to deliver the optimized solution information to the CR User according to the reasoning result gained using the classified case DB. To this end, in this paper artificial neural networks are used for learning and reasoning according to the characteristics of preprocessed history data based on the categorical data size and the use of data scaling techniques.

In CRN, ANN achieves a high prediction accuracy for future existing user patterns by learning the behavioral patterns of existing users from a data set [28,29,30]. The ANN structure consists of an input layer, a hidden layer, and an output layer. The input layer is responsible for delivering content image information composed of *N* neurons to the hidden layer. The hidden layer investigates whether a feature pattern is included in the content image received from the input layer and reports the content cost to the output layer. Assuming that the feature pattern is defined for each hidden layer, as illustrated in Figure 4, the calculation of the content cost for the content image included in the feature pattern is determined by the thickness (weight) of the connecting arrow between the hidden layer and the input layer, as shown in Figure 4. The output layer takes into account the content cost information calculated in the hidden layer and outputs the certainty as a value between zero and one. Finally, the final content image is judged by comparing the certainty factor of each output layer from ① to ⑨.

#### 2.3.1. Definition of Artificial Neural Network Parameters

In this section, variables necessary for describing the relationship between neurons constituting an ANN are defined and relational expressions for the content cost and certainty factor in the hidden layer and output layer in the previously constructed ANN are summarized [31,32].

First, when the *i*-th input layer neuron $x_{i}$ is given a weight $\omega_{i}^{H_{j}}$ and a threshold $\theta^{H_{j}}$, the output $y_{j}$ for each hidden layer is defined as σ(a).
(10)$$\sigma \left(a\right)=\frac{1}{1+e^{-a}}$$
where i∈N, N=1,2,3,…,n.

Here, σ(a) is a sigmoid function that is continuously expressed in the interval between zero and one, while the input linear sum *a* in the *j*-th hidden layer is defined as follows:(11)$$a=x_{1}{\omega_{1}}^{H_{j}}+x_{2}{\omega_{2}}^{H_{j}}+\ldots +\theta^{H_{j}}$$
where j∈M, M=1,2,3,…,m.

The role of the threshold value $\theta^{H_{j}}$ is to block the noise signal coming from transport sources other than the signal from the transport of the input layer connected by the thick arrow shown in Figure 4.

As described above, the content cost calculation result output by the sigmoid function determines the content image as an output certainty factor with a value between 0 and 1 for the output layer. For example, if the certainty factor of output layer ① is very close to 1, we can be sure that the input image has an average probability of occupation of 0.1.

#### 2.3.2. Defining Objective Functions for Weight and Threshold Optimization

In this study, linear regression analysis is used to derive the optimal weight and threshold values to minimize the sum of square errors $Q_{h}$ of the output layer for each piece of input data defined above.
(12)$$z_{k}^{target}=py_{j}+q$$

In the regression equation given above, $y_{j}$, $z_{k}^{target}$ are variables used for inputting the actual value of the data, the right-hand side $y_{j}$ is defined as the explanatory variable, and the left-hand side $z_{k}^{target}$ is defined as the objective variable. At this time, the square error $Q_{h}$ between the objective variable $z_{k}^{target}$ given to the *k*-th output layer and the predicted value $z_{k}^{trial}$ is calculated as follows:(13)$$Q_{h}=\sum_{k=1}^{l}{\left(z_{k}^{target}-z_{k}^{trial}\right)}^{2}=\sum_{k=1}^{l}{\left(z_{k}^{target}-\left(py_{j}+q\right)\right)}^{2}$$
where k∈L, k=1,2,3,…,l.

Here, the objective function $Q_{T}$ for optimizing the parameters $p(=\omega_{i}^{H_{j}})$ and $q(=\theta^{H_{j}})$ representing weights and thresholds can derive optimal parameters when the sum of $Q_{h}$ calculated from the total *T* input data is minimized.
(14)$$Q_{T}=\sum_{h=1}^{t}Q_{h}$$
where h∈T, T=1,2,3,…,t.

## 3. Case-Based Reasoning

In intelligent dynamic spectrum resource management, case-based reasoning (CBR) is divided into CR Mater and CR User roles as shown in Figure 5, and retrieving, reusing and retain operation based on the case DB created through data mining is performed [13,33].

### 3.1. CR Master

CR Master performs similarity verification using the optimal number of hidden layers and threshold that minimize the objective function $Q_{T}$ when a new case is input. If the predicted and actual values are the same during the similarity verification process, the CR Master will deliver a list of solutions and backup channels to the CR User; conversely, if they do not match the actual values, they will only deliver the backup channel list to the CR user.

### 3.2. CR User

The CR user must not interfere with incumbent users when a spectrum is shared opportunistically by identifying an idle channel, defined as an unused spectrum hole or white space, at a specific time and location. Therefore, if an incumbent user is detected during the idle channel reasoning process, the operation should be paused and switched to another idle channel to maintain communication. This switching method is defined as spectrum handoff and is used in this study to perform a reasoning engine operation utilizing a reactive handoff process based on the backup channel list. At this time, when a CR user using the maximum transmission power in the idle channel demonstrates an interference influence on incumbent users existing in an adjacent channel or location, the CR user performs solution reuse or an optimization engine operation depending on the similarity verification result. If the predicted and actual values are found to be the same after performing a similarity verification, the CR user reuses the solution received from the CR Master, we use Monte Carlo algorithm-based interference analysis to verify the solution.

In this study, interference analysis based on the Monte Carlo algorithm calculates randomly generated radio wave propagation-related samples based on input parameters applied to a specific interference scenario environment, determining the inter-user interference probability using the samples obtained [34]. Monte Carlo algorithm-based interference analysis is largely divided into user interfaces, event generation engines, and interference calculation engines. At first, the user interfaces define the parameters (antenna height, transmission power, center frequency, etc.) of the victim in the interference environment scenario, as well as a propagation loss model according to specific distance, location, and topography features. The interference environment is divided into victim and interfering links, as shown in Figure 6.

Victim links refer to incumbent users in use at a particular location and time in the licensing band, while interfering links refer to any communication that temporarily uses an idle channel not being used by the incumbent user at a specific time and region through the opportunistic frequency use method based on CR technology in the licensed band.

The event engine calculates the desired received signal strength (*dRSS*) and the interfered received signal strength (*iRSS*) through repeated simulations of *N* events according to the frequency separation and space distance. First, *dRSS* defines the transmit power ($P_{w}$) and antenna gain ($g_{w\to v}$,$g_{v\to w}$) for the desired transmitter and victim receiver constituting the victim link. It also uses a path loss model ($PL_{w\to v}$) that considers the distance between the wanted transmitter and the victim receiver in the city center as well as the propagation environment, which is defined by a number of unspecified obstacles.
(15)$$dRSS=P_{w}+g_{w\to v}-PL_{w\to v}+g_{v\to w}$$

$iRSS_{unwanted,i}$ is calculated based on the imperfect pulse shaping of the interfering source transmitter and propagation environment defined by the unwanted emission component ($emission(f_{It},f_{Vr})$) of the interfering source, as shown in Figure 7. This component appears in individual elements constituting the transmitter and a number of unspecified obstacles between the interfering transmitter and interfering receiver. Additionally, to obtain the sum of the signal powers received from *n* interfering transmitters operating in different channels and random locations, $iRSS_{unwanted,i}$ in dBm is used as an exponent and then converted into a logarithm, as defined in Equation (Equation 17).
(16)$$iRSS_{unwanted,i}=emission\left(f_{It},f_{Vr}\right)+g_{i\to v}-PL_{i\to v}+g_{v\to i}$$
where i∈N, N=1,2,3,…,n.
(17)$$iRSS_{unwanted,i}=10\log_{10}\left(\sum_{i=1}^{n}10^{\frac{iRSS_{unwanted,i}}{10}}\right)$$

Finally, according to the process of Figure 8, the interference engine calculates $C/I_{trial}$ using the iRSS value generated from the event with a dRSS value greater than the sensitivity and determines it to be “Good” or “Interfered” by making comparisons to the interference protection ratio ($C/I_{target}$) of the victim receiver. This process is repeated until the end of events, and after the events are terminated the interference probability of $N_{interfered}$, for which interference occurred among the total number of events, is finally derived.

As mentioned above, when verifying a solution using Monte Carlo algorithm–based interference analysis, the CR user determines the solution maintenance or optimization engine operation according to the 5% interference probability standard applied in the field test [35,36]. If the interference probability exceeds the standard of 5% or more during the solution verification process, the CR user determines that the solution information is not suitable and derives the optimized transmission parameters according to the service purpose required by the incumbent user through the optimization engine. Conversely, if the interference probability is within 5%, the CR user maintains the past solution.

Additionally, in the case where the predicted and actual values are different, the CR user determines that there has been no similar case in the past and directly derives the optimized transmission parameters according to the service purpose required by the incumbent user through the optimization engine.

## 4. Simulation

### 4.1. Scenario

In this study, the process of analyzing the performance of intelligent dynamic real-time spectrum resource management using data mining and case-based reasoning in a CR network environment that is dynamically variable in the space–time and frequency domains consists of three steps, as shown in Figure 9 below [13,18,37].

In the first step, historic data that are dynamically variable in the time and frequency domains are based on the super-frame structure defined in IEEE 802.22 Mac [38]. The historic data representing the super-frame consist of 16 time slots with a length of 10 ms. In addition, the channels are composed of 55 individual orthogonal sub-channels, each with a 100 kHz bandwidth size within the 5.5 MHz occupied bandwidth, excluding the 250 kHz protection area on both sides of the 6 MHz channel bandwidth defined in the ATSC broadcasting standard for domestic DTV broadcasting [39]. In this study, the number and distribution of incumbent users in the spatial domain are determined according to the channel usage status for each time slot unit using historic data composed of super-frame units, as described above [18].

The second step of forming *n* single wireless networks in the space domain is deploying wireless networks that opportunistically service IoT devices in random locations within a 5 × 5 km square area [40] using channels representing the incumbent user occupancy statuses of the 55 individual orthogonal sub-channels in the *m*-th time slot. At this time, in order to derive the optimal transmission parameter of the interfering transmitter that satisfies the standard of 6 dB for the protection ratio($I/N_{target}$) of the victim receiver at the minimum receiving power position, the IoT assumes the worst case situation located on the service boundary (600 m) [41] of a base station. Table 2 lists the victim link parameters used in the simulation.

As a final step, Figure 10 depicts a situation where 255 IoTs [38] perform uplink operations at any location within the service area (600 m) of a base station located in the center of a 5 × 5 km square area. This indicates that the unwanted emission of industrial IoT devices performing uplink operations with a maximum transmission power of 12.6 dBm per 100 kHz [39] at a random location can have a serious radio wave interference effect on incumbent users located in adjacent channels and cell boundaries [42,43]. In addition, the extended Hata propagation model [34] and the unwanted emission mask of the wireless microphone service [44] are applied to reflect the propagation loss due to an unspecified obstacle between the interfering source and receiving unit. Table 3 and Table 4 show the interfering link parameters and unwanted emission masks used for the simulation.

### 4.2. Performance Evaluation Method

Similarity verification is performed using the optimal number of hidden layers, weights, and thresholds to minimize the objective function $Q_{T}$ based on the preprocessed training data. At this time, the accuracy is calculated by counting the number of successful predictions in the total *T* test data sets by producing a value of one if $z_{k}^{target}$ and $z_{k}^{trial}$ are equal in the *h*-th test dataset and a value of zero otherwise.
(18)$$F\left(Q_{h}\right)=\left\{\begin{array}{cc} 1, & z_{k}^{trial}=z_{k}^{target}\\ 0, & \mathrm{otherwise} \end{array}\right.$$
(19)$$Accuracy=\frac{1}{T}\sum_{h=1}^{t}F\left(Q_{h}\right)$$

In addition, to check the number of optimization engine operations according to the prediction accuracy, the following conditional expression is defined:

If the predicted value $z_{k}^{target}$ and the actual value $z_{k}^{trial}$ are the same, CR Users reuse the solution when interference occurs in the reasoning engine process. At this time, the verification for the solution is defined as zero when the interference probability criterion ($P_{IP,target}$) of 5% is satisfied through the Monte Carlo algorithm-based interference analysis and when the interference probability exceeds 5%, it is defined as 1 because the optimization engine operation is performed.
(20)$$F_{1}\left(T_{i}^{h}\right)=\left\{\begin{array}{cc} 1, & P_{IP,trial}>P_{IP,target}\\ 0, & P_{IP,trial}<P_{IP,target} \end{array}\right.$$

Conversely, if the predicted value differs from the actual value, the number is defined as zero if the interference probability criterion of 5% is satisfied when idle channels are used at maximum transmission power in the reasoning engine process and one when it exceeds 5%.
(21)$$F_{2}\left(T_{i}^{h}\right)=\left\{\begin{array}{cc} 1, & P_{IP,trial}>P_{IP,target}\\ 0, & P_{IP,trial}<P_{IP,target} \end{array}\right.$$

Following these conditions, the number of optimization engine operations per time slot unit $T_{i}$ is calculated according to the prediction accuracy during *T* test data sets.
(22)$$Count\left(T_{i}^{h}\right)=\sum_{h=1}^{t}\sum_{i=1}^{m}\left(F_{1}\left(T_{i}^{h}\right)+F_{2}\left(T_{i}^{h}\right)\right)$$

The definitions of the number of spectrum handoffs, handoff delay, link maintenance probability, and energy consumption are defined as follows for intelligent dynamic spectrum resource management performance comparison and analysis depending on whether data mining and case-based reasoning is applied or not.

The number of spectrum handoffs is defined as the number of times that data transmission is stopped and switched to another idle channel according to the incumbent user detection during one session of CR data transmission in the idle channel reasoning process [13,45,46]. However, as frequent spectrum handoff operations can degrade the communication performance based on latency and increase the energy consumption, this simulation aims to minimize unnecessary spectrum handoffs in order to realize an efficient CR network. The number of spectrum handoff operations was counted for each time slot according to the incumbent user detection as defined in the following equation:(23)$$F\left(T_{i}\right)=\left\{\begin{array}{cc} 1, & \mathrm{if}\;T_{i}\;\mathrm{is}\;‘\mathrm{on}\;\mathrm{state}’\\ 0, & \mathrm{otherwise}, \end{array}\right.$$
(24)$$Count\left(T_{i}\right)=\sum_{i=1}^{m}F\left(T_{i}\right)$$

The spectrum handoff latency represents the number of channel searches that occur during the handoff process to an idle channel according to the detection of an incumbent user [13,45,47]. To verify a channel selection technique that can quickly handle an idle channel with a small number of searches during the reasoning period, as shown in Equations (Equation 25) and (Equation 26), the number of channel searches occurring during the handoff process for each time slot is counted according to the backup channel order:(25)$$F\left(C_{ij}\right)=\left\{\begin{array}{cc} 1, & \mathrm{if}\;C_{ij}\;\mathrm{is}\;‘\mathrm{on}\;\mathrm{state}’\\ 0, & \mathrm{otherwise}, \end{array}\right.$$
(26)$$Count\left(C_{ij}\right)=\sum_{i=1}^{m}\sum_{j=1}^{n}F\left(C_{ij}\right)$$

In a temporally and spatially dynamic CR network, frequent spectrum handoffs made by incumbent users can degrade IoT communication performance as a result of excessive sensing energy consumption [13,48]. Therefore, in this study, as shown in Equations (Equation 27) and (Equation 28), a reduction in sensing energy consumption was confirmed by counting the number of times the same idle channel was maintained in a continuous time interval without a spectrum handoff operation occurring over a total of m−1 time slots.
(27)$$F\left(T_{i},T_{i+1}\right)=\left\{\begin{array}{cc} 1, & \mathrm{if}\;T_{i}\;\mathrm{and}\;T_{i+1}\;\mathrm{is}\;‘\mathrm{off}\;\mathrm{state}’\\ 0, & \mathrm{otherwise} \end{array}\right.$$
(28)$$Count\left(T_{i},T_{i+1}\right)=\sum_{i=1}^{m-1}F\left(T_{i},T_{i+1}\right)$$

A pause in communication is most likely to occur when an idle channel does not exist at a specific time in a traffic environment with a large number of incumbent users. Additionally, radio wave interference caused by the unwanted emission of an interference source limits the use of available idle channels [13,46,49]. Therefore, to evaluate the spectrum management performance for continuous communications in such a scenario, the number of time slots in which a communication link was successfully maintained up to $T_{m}$ was counted.
(29)$$F\left(T_{i}\right)=\left\{\begin{array}{cc} 1, & \mathrm{if}\;T_{i}\;\mathrm{is}\;‘\mathrm{CU}\;\mathrm{is}\;\mathrm{on}\;\mathrm{state}’\\ 0, & \mathrm{otherwise} \end{array}\right.$$
(30)$$Count\left(T_{i}\right)=\sum_{i=1}^{m}F\left(T_{i}\right)$$

### 4.3. Performance Evaluation Results

#### 4.3.1. Data Preprocessing Using History Data

The data preprocessing process, which is conducted using historic data, calculates the frequency of occurrence of occupation probability values per channel using 5, 10, and 20 equal interval sizes to convert the statistical data for each of the nine average occupancy probabilities composed of 0.1 size units from 0.1 to 0.9 into categorical data. Figure 11, Figure 12 and Figure 13 show the histogram represented by the average occupancy probability of categorical data according to the equal interval size and the pattern by data characteristics according to discretization and normalization.

In cases where $P_{ON}^{AVG}$ = 0.1, the probability density function has a long tail on the right side, indicating that more data are distributed on the left side, meaning that out of the 55 channels the number of idle channels available to CR users is larger. Conversely, if $P_{ON}^{AVG}$ = 0.9, the number of channels with a high occupancy probability of incumbent users is distributed to the right, so the number of idle channels available to CR users is small. Therefore, it can be seen that the distribution of data gradually shifts from left to right as the average occupancy probability increases.

In the case of 4×5, 2×5, and 1×5 images pretreated with discretization according to the unit of 5, 10, and 20, as the equidistant size gradually increases, it is highly likely that they will be misclassified between adjacent cases as shown in Figure 11.

On the other hand, when pretreated through normalization, it is easy to distinguish the characteristics of each case under conditions such as discretization. However, if the equidistant size is very low or high, it is likely to be misclassified as the presence of neurons with similar appearance frequencies. Therefore, in this paper, a preprocessing process through categorical data with an appropriate equidistant size is required.

#### 4.3.2. Comparative Analysis of Prediction Accuracy by Data Preprocessing Techniques, Number of Hidden Layers, and Number of Learning Data Samples Using History Data

Figure 14, Figure 15 and Figure 16 compare and analyze the prediction accuracy according to the number of training data (1, 5, 10, 50, 100, 200, and 300) and the number of hidden layers (2, 3, 4, and 5) for each case using preprocessed learning and test data that have undergone discretization and normalization with a categorical data size (m) set to 5, 10, and 20. Overall, it shows a graph that exponentially increases along with the size of the training data while the number of hidden layers gradually increases. At this time, it can be seen that when the training and test data preprocessed by the discretization method are used, the accuracy improves as the size of the categorical data set increases.

This requires categorical data with a high resolution to improve the accuracy, since the prediction accuracy is evaluated to be low due to misclassification caused by an unclear distinction from other adjacent cases when data with a relatively low resolution are used. On the other hand, in the case of data preprocessing using a normalization method under the same conditions, the accuracy is high even with a low-resolution categorical data size. Therefore, it can be seen that if the categorical data size is five, the prediction accuracy can be improved by approximately 38% based on the maximum value and there will be a 19% and 11% higher prediction accuracy for the other 10 and 20, respectively.

In the case of prediction accuracy according to the number of learning data available, it can be seen that the growth trend begins to clearly slow from having 50 learning data samples in common and converge at 100 learning data samples without a significant increase. Therefore, from the results gained in this study, we can state that 100 learning data points is the optimal number of learning data necessary to improve the prediction accuracy and minimize the increase in computation due to the excessive amount of learning data.

In addition, when analyzing the prediction accuracy using the same number of learning data, it can be seen that as the number of hidden layers increased, the classification ability improved and more patterns could be recognized, indicating that the prediction accuracy increased. However, because the amount of computation necessary may increase due to overfitting, four hidden layers (green) were used in this study as the optimal number of hidden layers with a high prediction accuracy while minimizing the amount of computation needed through the above simulation results. This has the same prediction accuracy as the maximum number of hidden layers in the 100 learning data samples.

In addition, when the hyperbolic tangent function [32] was used under the same conditions as the artificial neural network structure optimized through the sigmoid function, the growth trend began to gradually decline in 100 training data as shown in Figure 17, and 84.71% prediction accuracy was obtained from 300 training data. That is, since the hyperbolic tangent function has about 13% lower prediction accuracy than the sigmoid function and requires more than 300 training data for high prediction accuracy, an optimization method using the sigmoid function is preferable.

#### 4.3.3. Comparison and Analysis of the Number of Optimization Engine Operations According to the Prediction Accuracy for the Number of IoT Devices

This section analyzes the number of optimization engine operations according to the prediction accuracy for each number of preprocessed learning data during pattern classification and case-based reasoning based on an ANN with four hidden layers.

At this time, when the predicted value and the actual value are found to be the same when verifying the similarity in the case-based reasoning process, the solution information delivered from the case DB is defined in Table 5 for each number of IoT devices. For reference, Table 5 shows the average values of transmission power optimized through reasoning engines and optimization engines using the number of IoT devices and 500 historic data points for each case.

As shown in Figure 18 below, the number of optimization engine operations according to the prediction accuracy for each number of training data are shown in a graph in which the number of optimization engine operations exponentially decreases as the accuracy increases.

At this time, the number of optimization engine operations using 255 IoT devices showed up to 1153 optimization engine operations at a 66.82% prediction accuracy, and it is possible to confirm that the optimization engine operation reduced to 833 for a total of 100 training data points with a 97.13% prediction accuracy. when comparing the number of optimization engine operations for 100, 50, and 10 IoT devices based on the same prediction accuracy as that given above, it can be seen that the number of optimization engine operations decreased slightly to 802, 1014, and 736, respectively.

In addition, as the number of IoT devices decreases, the number of idle channels available through the optimization engine operation increases. However, for the case of 10 IoT devices, the number of optimization engine operations is lower than the number of 50 IoT devices, as there are some idle channels that can be used at maximum transmission power without the operation of the optimization engine.

#### 4.3.4. Comparative Analysis of the Number of Optimization Engine Operations Depending on Whether or Not Case-Based Reasoning Is Applied

In Figure 19, the red bar graph represents the number of optimization engine operations that occur when interference is affected by incumbent users in adjacent channels and locations while reasoning an idle channel based on a list of backup channels for every time slot through the existing intelligent dynamic spectrum resource management. On the other hand, the blue bar graph shows the number of optimization engine operations for each number of IoT devices out of 100 training data with a 97.4% prediction accuracy in case-based reasoning based on optimized data mining. That is, when case-based reasoning is applied based on optimized data mining, it can be seen that the optimization engine operation is significantly reduced by more than half compared to the existing one for all traffic situations from 0.1 to 0.9.

In addition, when 255 IoT devices are used, as the average occupancy probability gradually increases, the frequency of the occurrence of incumbent users in adjacent channels and areas increases, resulting in optimization engines operating in up to 0.5 traffic.

However, frequent handoff operations occurring due to the frequent appearance of incumbent users from 0.6 traffic increases the number of channel searches, making it difficult to maintain a backup channel list. Eventually, a pause begins to occur at certain time slots and the number of idle channels available gradually decreases, resulting in a sharp decrease in the number of optimization engine operations.

On the other hand, in cases where there are 100, 50, and 10 IoT devices, the use of a relatively small number of devices will increase the number of idle channels that can be used as optimized transmission parameters, even in traffic situations with a high occupancy probability. Therefore, it can be seen that the optimization engine operation increases up to a level of 0.8 traffic.

#### 4.3.5. Comparative Analysis of Spectrum Resource Management Performance Depending on Whether or Not Case-Based Reasoning Is Applied

In the case of spectrum resource management using 10 IoT devices, as shown in Figure 20, continuous handoff operation is possible in situations where there is up to 0.9 traffic. In addition, compared to the performance of 255 IoT devices when the number of channel searches was less than 500 and the number of time slots that could be used continuously had increased by more than seven, it was confirmed that the link maintenance performance was improved by more than 90%, which is very similar to the value of the existing intelligent dynamic spectrum resource management performance

In the case where there are 50 IoT devices, there is a continuous handoff operation in up to 0.8 traffic and a channel search number of approximately 345 or less. In addition, more than three time slots are continuously available on average and the link maintenance performance improves by more than 65% compared to that gained with 255 IoT devices; this is very similar to the performance of existing intelligent dynamic spectrum resource management.

Finally, in the case where there are 100 IoT devices, there is a continuous handoff operation in up to 0.6 traffic and a number of channel searches of approximately 270 or less. In addition, more than three time slots are continuously available on average, and it can be seen that the link maintenance performance improves by approximately 20% or more compared to that gained with 255 IoT devices; this is very similar to the performance of existing intelligent dynamic spectrum resource management.

## 5. Conclusions

In this paper, we proposed a method for intelligent dynamic real-time spectrum resource management using data mining and case-based reasoning to overcome the complexity of existing methods and the difficulty of rapid resource allocation.

Data mining facilitated classification according to historic data characteristics through data preprocessing using historic data and learning through ANN. Additionally, the reuse of solutions through case-based reasoning reduced the complexity of the existing intelligent dynamic spectrum resource management and enabled efficient real-time resource allocation. also, the fact that our method achieved the same number of spectrum handoffs and the same handoff delay, energy consumption, and link maintenance probability as the existing spectrum resource management performance method proved that reliable communication can be maintained at every time slot without optimization engine operation.

As such, the intelligent dynamic real-time spectrum resource management proposed in this study is not only used in various fields such as smart factories, smart farms, and autonomous vehicles that require real-time processing, but also By realizing frequency sharing, it will be possible to respond to excess spectrum demand and promote the growth of new services.

However, in addition to the occupancy probability by channel in the ON/OFF statistical model considered in this paper, there are numerous factors that can affect learning performance, such as service priorities, locations, and QoS requirements from a CR User perspective. Therefore, future studies need to prepare reasonable integration measures for more accurate estimation.

## Figures and Tables

**Figure 1 sensors-21-07902-f001:**
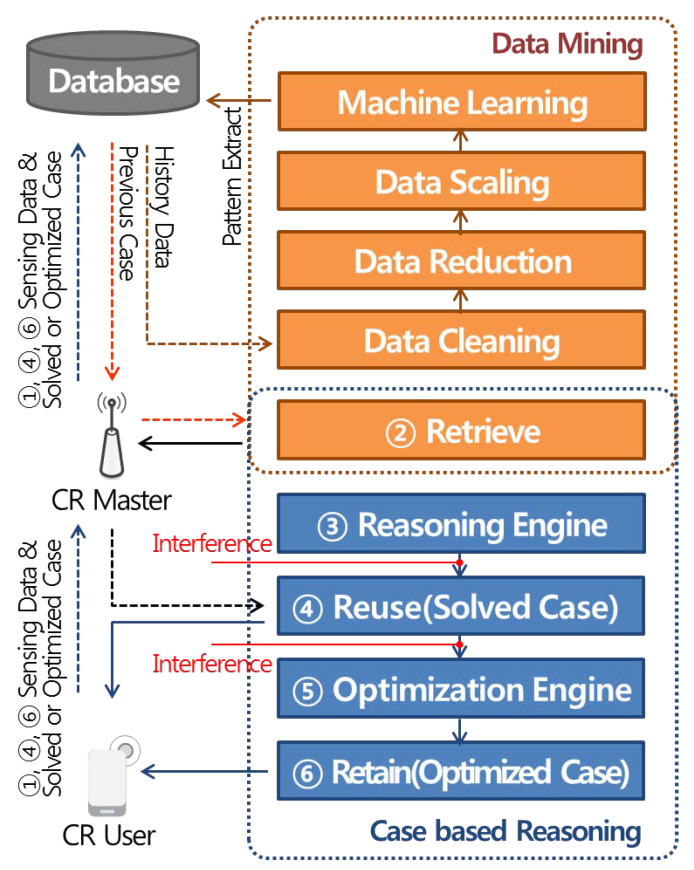
Intelligent dynamic real-time spectrum resource management structure based on data mining and case-based reasoning.

**Figure 2 sensors-21-07902-f002:**
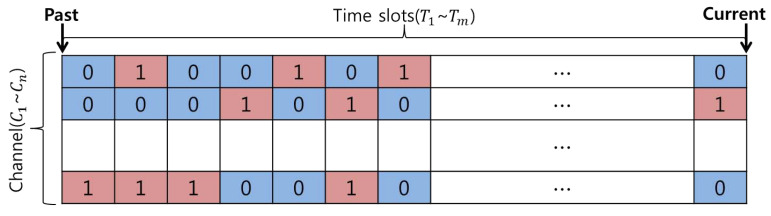
Incumbent user activity modeling based on an ON/OFF model.

**Figure 3 sensors-21-07902-f003:**
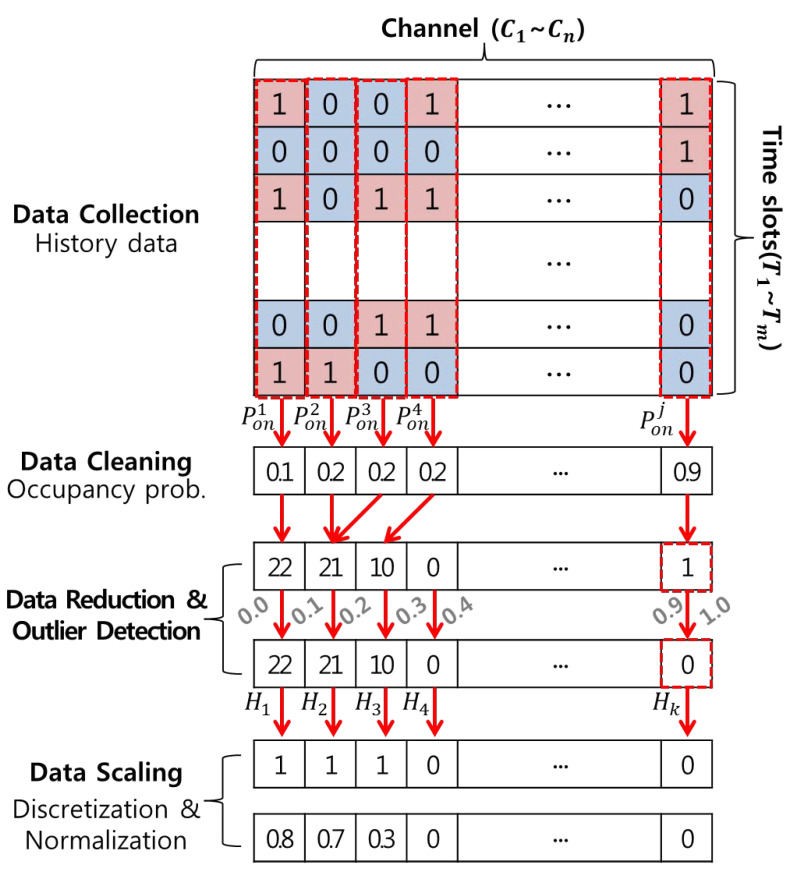
Data preprocessing using historic data.

**Figure 4 sensors-21-07902-f004:**
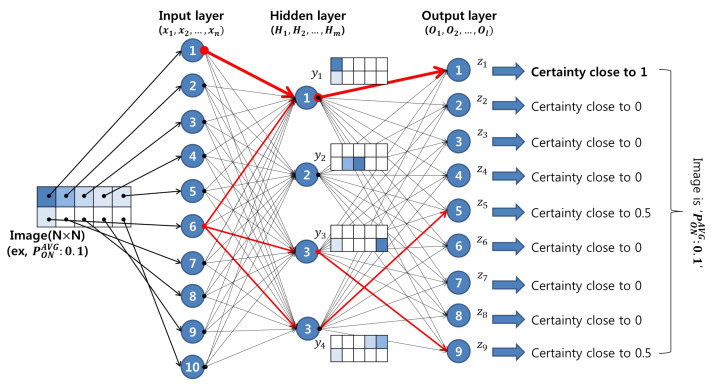
Artificial neural network structure.

**Figure 5 sensors-21-07902-f005:**
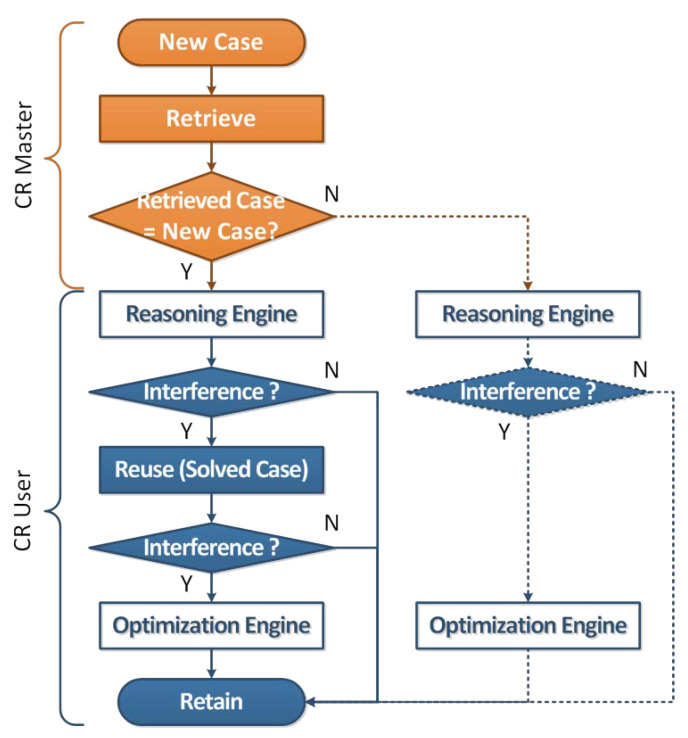
Flowchart of the intelligent dynamic real-time spectrum resource management using case-based reasoning.

**Figure 6 sensors-21-07902-f006:**
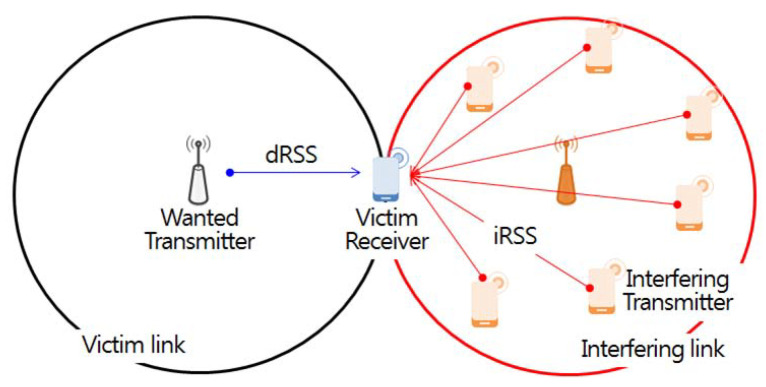
Victim link and interfering link.

**Figure 7 sensors-21-07902-f007:**
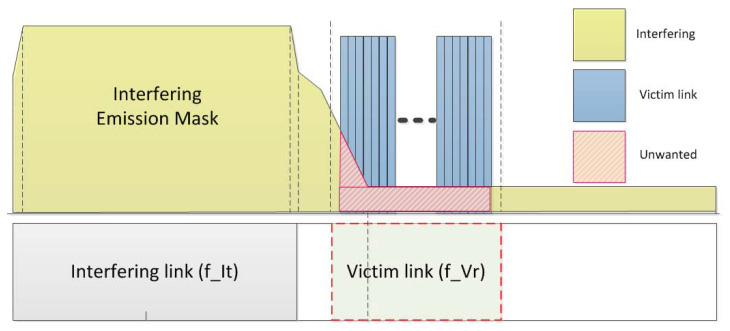
Interference caused by unwanted emission masks.

**Figure 8 sensors-21-07902-f008:**
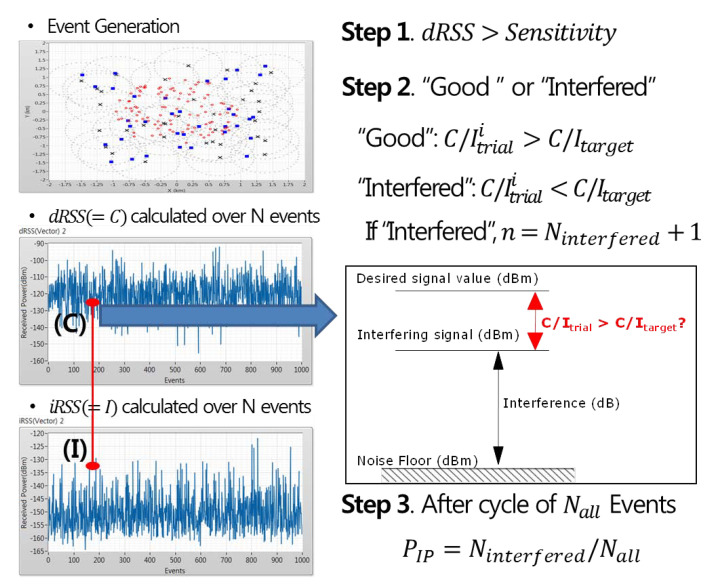
Interference probability calculation.

**Figure 9 sensors-21-07902-f009:**
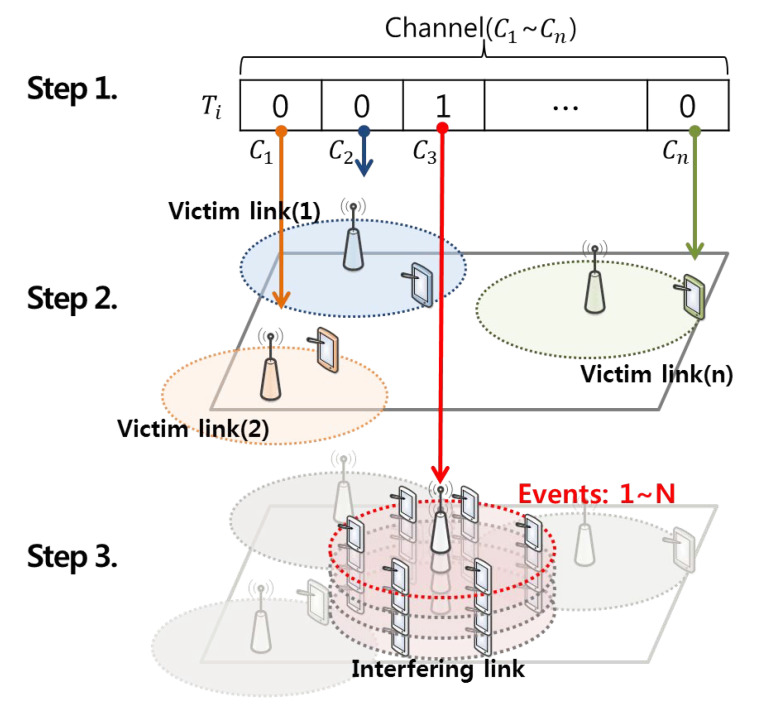
Simulation environment created by the space–time and frequency domains.

**Figure 10 sensors-21-07902-f010:**
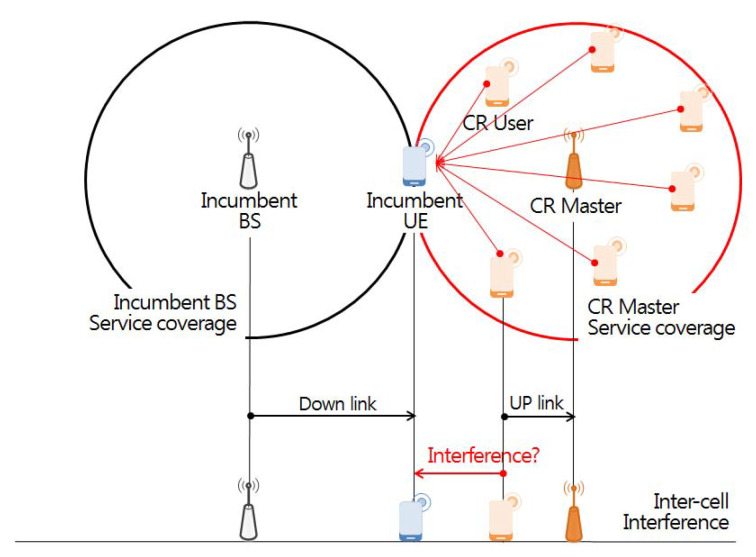
Inter cell interference scenario.

**Figure 11 sensors-21-07902-f011:**
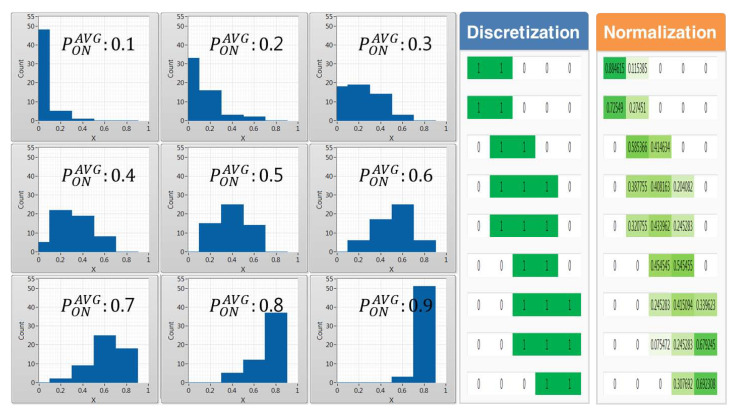
(Δx = 20, *L* = 5) Histogram by average occupancy probability and pattern comparison by the data scaling method.

**Figure 12 sensors-21-07902-f012:**
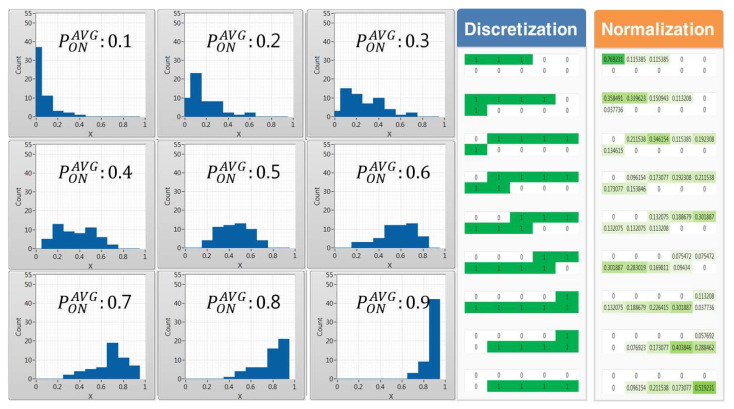
(Δx = 10, *L* = 10) Histogram by average occupancy probability and pattern comparison by the data scaling method.

**Figure 13 sensors-21-07902-f013:**
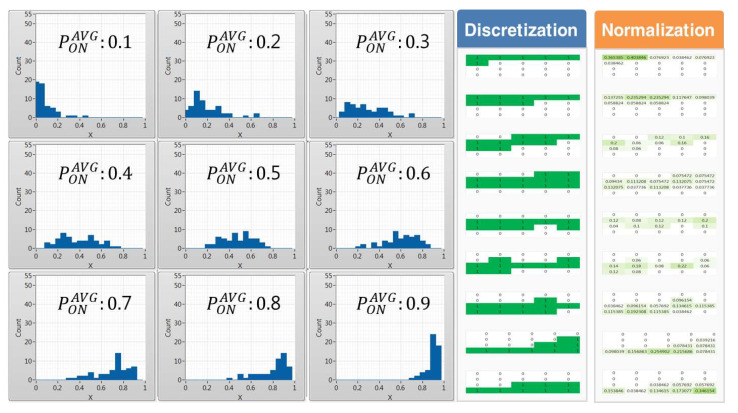
(Δx = 5, *L* = 20) Histogram by average occupancy probability and pattern comparison by the data scaling method.

**Figure 14 sensors-21-07902-f014:**
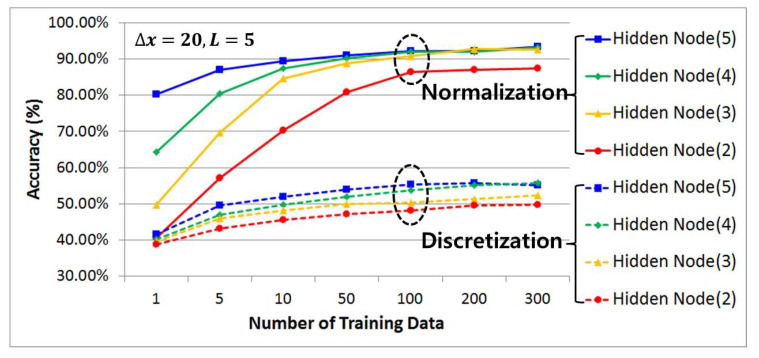
(Δx = 20, *L* = 5) Comparative analysis of prediction accuracy per data preprocessing technique using historic data, the number of hidden layers, and the number of training data.

**Figure 15 sensors-21-07902-f015:**
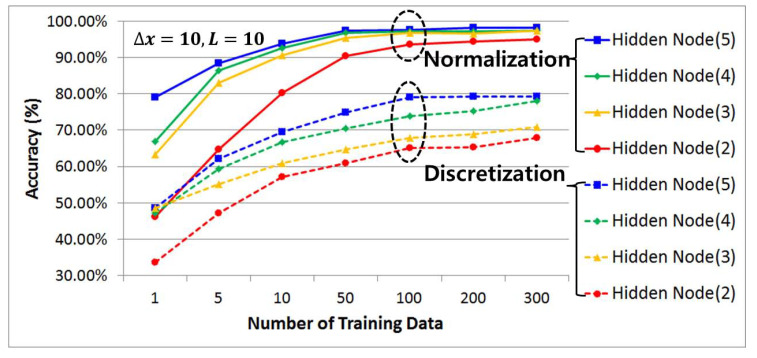
(Δx = 10, *L* = 10) Comparative analysis of prediction accuracy per data preprocessing technique using historic data, the number of hidden layers, and the number of training data.

**Figure 16 sensors-21-07902-f016:**
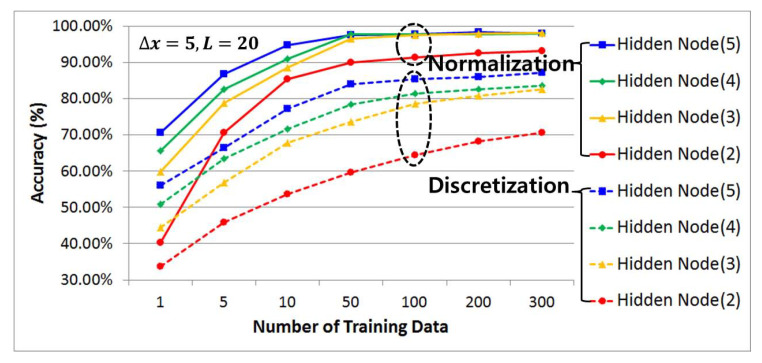
(Δx = 5, *L* = 20) Comparative analysis of prediction accuracy per data preprocessing technique using historic data, the number of hidden layers, and the number of training data.

**Figure 17 sensors-21-07902-f017:**
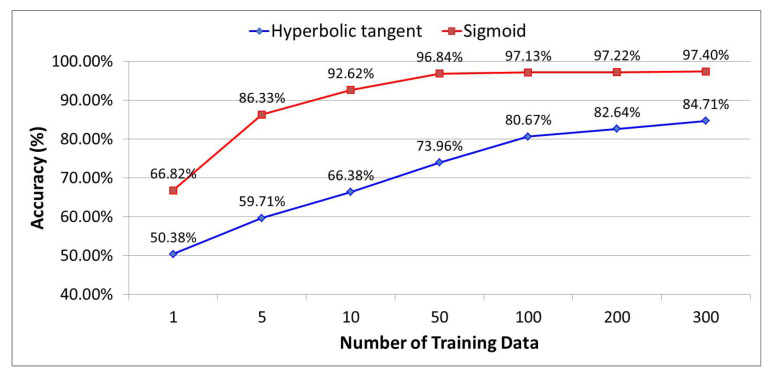
Comparative analysis of prediction accuracy between sigmoid function and hyperbolic tangent function.

**Figure 18 sensors-21-07902-f018:**
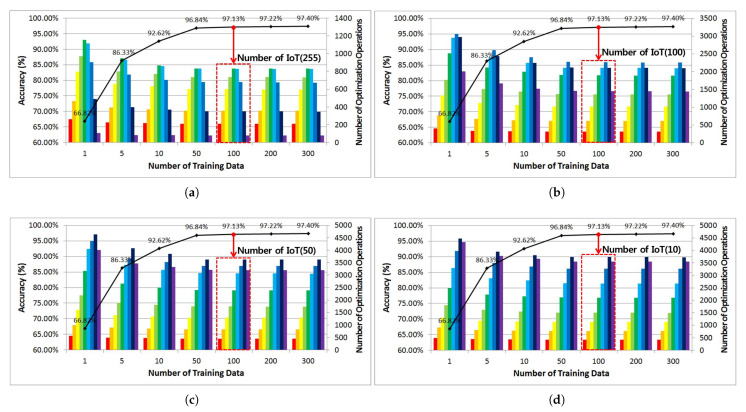
Comparison and analysis of the number of optimization engine operations according to the prediction accuracy for the number of IoT devices. (**a**) Number of IoT: 255, (**b**) Number of IoT: 100, (**c**) Number of IoT: 50, (**d**) Number of IoT: 10.

**Figure 19 sensors-21-07902-f019:**
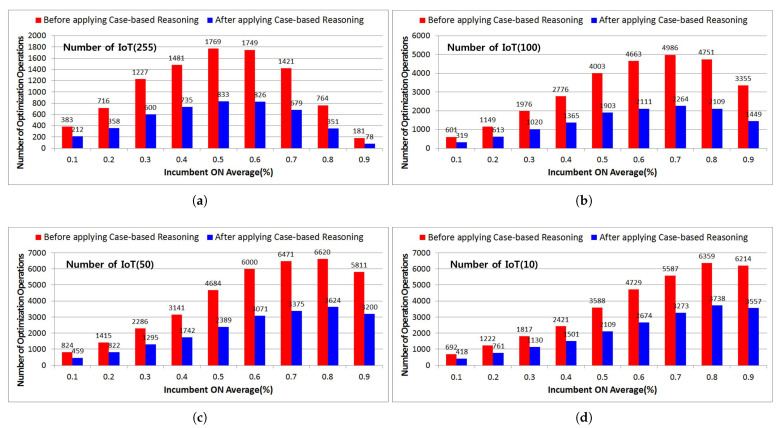
Comparative analysis of the number of optimization engine operations depending on whether or not case-based reasoning is applied. (**a**) Number of IoT: 255, (**b**) Number of IoT: 100, (**c**) Number of IoT: 50, (**d**) Number of IoT: 10.

**Figure 20 sensors-21-07902-f020:**
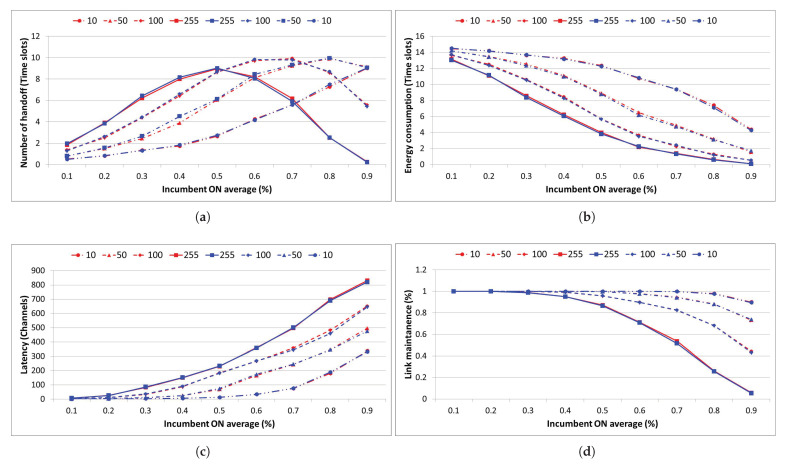
Comparative analysis of spectrum resource management performance ((**a**) spectrum handoff, (**b**) energy consumption, (**c**) latency, (**d**) link maintenance) depending on whether or not case-based reasoning is applied.

**Table 1 sensors-21-07902-t001:** Technologies driving spectrum demand.

Technology	Spectrum Demand
5G	-5G mobile networks offer significant potential to increase data transfer capacity, as well as spectrum efficiency.
	-Sub-6 GHz bands have relatively better propagation characteristics, offering a wider coverage area than mmWave, but their heavy incumbent use limits large, contiguous spectrum blocks.
	-mmWave bands offer more spectrum due to less incumbent use, allowing for wider bandwidths, supporting higher throughputs. However, its use is limited, by lower propagation characteristics making them more suitable for coverage of relatively small areas, usually in dense environments.
IoT	-Increased connectivity and capacity introduced by technologies using licensed and unlicensed spectrum are fostering more connected devices as part of IoT ecosystem.
	-Successful consumer and public applications of different IoT technologies are reliant on effective and efficient spectrum management.
	-Spectrum requirements for various segments of the IoT landscape depend on user cases specific to their application. for example, connections for use by industrial automated robots are more latency sensitive than connected kitchen appliances.
WiFi	-Wireless network technologies are critical to connected devices implementation and IoT ecosystem advancement.
	-In addition to previous use of 900 MHz, 2.4 GHz, and 5 GHz bands, newer WiFi technologies are being implemented in 60 GHz (57–66 GHz) and 6 GHz (5925–7125 MHz) bands.
	-Several countries (e.g., United States and United Kingdom) are increasing availability of 6GHz band for unlicensed use.
	-New rules in the United States make available 1200 MHz of the spectrum for unlicensed use in the 6 GHz band.
HAPS	-HAPS applications (i.e., radio stations located in the stratosphere between 20 and 50 km above the Earth’s surface) can expand access to wireless connectivity.
	-HAPS support other terrestrial technologies with potential to expand connectivity and telecommunications services in rural and remote areas.
	-HAPS can serve as a tool to extend existing terrestrial networks and provide higher quality service to already connected areas as well as connectivity during emergency situations.
	-HAPS applications have frequency bands either authorized directly to its provider or to an existing partner telecommunications operator, such as a mobile operator.
NGSO	-NGSO satellites systems, comprised of hundreds or even thousands of satellites, provide connectivity in area currently unreached by terrestrial telecommunications infrastructure.
	-This presents some spectrum management challenges, in terms of managing use of different frequency bands and allowing GSO and NGSO satellite systems to operate simultaneously, while mitigating the risk of harmful interference.

**Table 2 sensors-21-07902-t002:** Victim link parameters.

Parameters	Value
Victim receiver center frequency ($f_{Vr}$)	600.2 MHz
Antenna height ($H_{w,v}$)	1.5 m
Antenna gain ($g_{w\to v}$)	6 dBi
Noise floor ($N_{f}$)	−167.83 dBm
Bandwidth (*B*)	100 kHz
Protection ratio ($I/N_{target}$)	−6 dB

**Table 3 sensors-21-07902-t003:** Interfering link parameters.

Parameters	Value
Interfering transmitter center frequency ($f_{It}$)	Among 55 sub-channels with a bandwidth of 100 kHz, the idle channel selected based on the backup channel is defined as the center frequency.
transmit power ($P_{It}$)	-Optimal transmission power that satisfies the criteria for interference probability within 5%;
	-(Min) 1 dBm (Max) 12.6 dBm.
Antenna height ($H_{i,v}$)	1.5 m.
Antenna gain ($g_{i\to v}$)	6 dBi.
Path loss ($PL_{i\to v}$)	Extended Hata model.

**Table 4 sensors-21-07902-t004:** Break points of the unwanted emission mask (B: bandwidth).

Frequency Relative to the Center of the Channel	Relative Level (dBc)
$f_{c}$ ± 1 MHz	−90
$f_{c}$ ± B	−80
$f_{c}$ ± 0.5 B	−60
$f_{c}$ ± 0.35 B	−20
$f_{c}$ ± 0.25 B	0

**Table 5 sensors-21-07902-t005:** Optimized transmission power according to the number of IoT devices.

Parameter	Number of IoT
Solution($P_{It}$)	(255) 6.54844 dBm, (100) 5.11569 dBm, (50) 4.65692 dBm, (10) 4.76417 dBm

## Data Availability

Not applicable.

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
