# Peer review of "Intelligent Dynamic Real-Time Spectrum Resource Management for Industrial IoT in Edge Computing"

_sensors, 2021, doi:10.3390/s21237902_

Round 1
Reviewer 1 Report
The idea is interesting, however, the addition of ANN is not properly explained.
Section 2.3 to 3 presents well-known concepts from ML and MLP (Multi-Layer Perceptron, the type of ANN used on the paper). Fig. 4 present a typical MLP topology, however, the input is an image with NxN entries (although the figure represents a 4x3 image) that was little correlation with the model used in the evalution.
Given the current trend with deep learning (with more advanced topologies), and the added bonus that the pre-processing can be reduced, there is little novelty in this work as it is presented.
Reviewer 2 Report
The article deals with the problem of the "intelligent" dynamic real-time resource management approach applicable in IoT networks. The merit can be interesting for computer scientists and other specialists in data mining. However, the form of the article practically excludes a broader audience - for example, the users of IoT technologies from different disciplines. I understand that the merit is very specialist and dedicated to solving particular problems, however, I am convinced that the journal scope is also committed to another group of specialists using different sensors. Therefore, I would suggest providing readers with some motivations for undertaking that particular kind of study, their usefulness and applicability.
My concern also inspires numerous formulas used in the text. Due to the fact the relevant references barely support them, I do not know whether they belong to the authors' findings, modifications of the existing formulas and laws or something else. Please provide necessary reference citations or explain the copyrights.
Using words like "intelligent", "smart", etc. to artificial algorithms is always disputable. An algorithm itself cannot be intelligent because intelligence is a property of humans. Would you please provide readers with the necessary explanation about that (I would recommend changing the phraseology)?
Finally, I would suggest concluding with giving some examples of practical aspects of the authors' studies.
Should the authors provide necessary changes to their manuscript, it can be resubmitted for another reviewing process.
Reviewer 3 Report
REVIEW
Article titled: “Intelligent Dynamic Real-Time Spectrum Resource Management for Industrial IoT in Edge Computing”
Sensors no. 1447756
List of Authors:
Deok-Won Yun, Won-Cheol Lee
- In this article, the Authors propose intelligent dynamic real-time spectrum resource management by applying data mining and case-based reasoning, which reduces the complexity of existing intelligent dynamic spectrum resource management and enables efficient real-time resource allocation. As the Authors claim, data mining and case-based reasoning analyzes the activity patterns of incumbent users using vast amounts of sensing data from industrial IoT and enables rapid resource allocation, using case DB classified by case.
- In Introduction, the Authors made extensive literature review dealing with the most important aspects about benefits of positioning-aided communication technology in high-frequency industrial IoT, access management techniques in machine-type communications, intelligent dynamic spectrum resource management based on sensing data in space-time and frequency domain, spectrum inference in cognitive radio networks, data preprocessing techniques toward efficient and reliable knowledge discovery from building operational data, ANN based spectrum inference for occupancy prediction in cognitive radio networks, and many other very interesting articles.
Bearing in mind a more efficient Spectrum Source Management very important is “the information superiority” and the sophisticated process of building the decision-making process in Database not only in the civil world but also in the military sense.
So, in this way the following articles entitled “Data Fusion in the Decision-making Process Based on Artificial Neural Networks” – Silesian Univ. of Technology, z.149, pp.97-108, should be listed in references list.
- A number of optimization engine operations are confirmed, as the Authors claim. But nowhere in this article, I haven't found the optimization criteria. This issue should be clarified.
- The obtained (by Authors) indicators prove that it is possible to minimize the complexity of existing intelligent dynamic spectrum resource management and maintain efficient real-time resource allocation and reliable communication.
How do the Authors of this article understand the definition of "real-time"?
- In Section 2.3 entitled “Machine Learning” the structure of ANN is shown (Figure 4). So my question is:
What would be the behavior of an Artificial Neural Network if another activation function was used instead of a sigmoid activation function?
The article is very interesting, but it requires systematization of the concepts/definition and answers to the questions contained in the review of this article.
The Authors addressed a problem which is relevant and appealing for this journal. However, I cannot recommend the current manuscript for publication unless the current version is corrected. After providing the amendments to the article, the work ought to be reviewed once again.

Reviewer 4 Report
The authors should consider the following minor comments to improve the paper:
- Figures 1 and 5 should be improved with a clear description of the components. Also, the sizes of the content should be minimised according to normal text.
- The contributions are not clearly presented in the abstract and introduction. "In this study, we confirmed" - proposed/introduced/defined would be better wording according to the nature of contributions.
- Please include a bullet-points contributions in the second last paragraph (Introduction). The last paragraph should be seen as an outline of the paper.
Round 2
Reviewer 1 Report
In my last review, I pointed out the following concern: "Given the current trend with deep learning(with more advanced topologies), and the added bonus that the pre-processing can be reduced, there is little novelty in this work as it is presented."
I understand your reply, but please include it in the manuscript.
Reviewer 2 Report
The new manuscript I got for reviewing has been improved. The authors introduced necessary changes as well as they have followed my suggestions. In my opinion, the text is much clearer now and can proceed with publishing. However, I recommend to the authors providing thorough proofreading because I still spotted some minor typing errors!
Kind regards,
Author Response
Please see the attachment.
We sincerely thank the reviewer for their good evaluation.

Reviewer 3 Report
REVIEW_2
Article titled: “Intelligent Dynamic Real-Time Spectrum Resource Management for Industrial IoT in Edge Computing”
Sensors no. 1447756
List of Authors:
Deok-Won Yun, Won-Cheol Lee
Zhiting Fei, Jiachen Zhao , Zhe Geng, Xiaohua Zhu, Jindong Zhang
The article Sensors no. 1447756 entitled “Intelligent Dynamic Real-Time Spectrum Resource Management for Industrial IoT in Edge Computing” has been carefully modified and well revised. The work is supposed to be finally accepted for publication in Sensors.

Author Response
We sincerely thank the reviewer for their good evaluation.